# Stochastic Thermodynamics of Multiple Co-Evolving Systems—Beyond Multipartite Processes

**DOI:** 10.3390/e25071078

**Published:** 2023-07-17

**Authors:** Farita Tasnim, David H. Wolpert

**Affiliations:** 1Massachusetts Institute of Technology, Cambridge, MA 02139, USA; farita@mit.edu; 2Santa Fe Institute, Santa Fe, NM 87501, USA; 3Complexity Science Hub, Josefstadter Straße 39, 1080 Vienna, Austria; 4Center for Bio-Social Complex Systems, Arizona State University, Tempe, AZ 85287, USA; 5International Center for Theoretical Physics, 34151 Trieste, Italy

**Keywords:** stochastic thermodynamics, thermodynamic uncertainty relations, thermodynamic speed limit theorems, composite processes, mismatch cost, periodic processes, multipartite processes

## Abstract

Many dynamical systems consist of multiple, co-evolving subsystems (i.e., they have multiple degrees of freedom). Often, the dynamics of one or more of these subsystems will not directly depend on the state of some other subsystems, resulting in a network of dependencies governing the dynamics. How does this dependency network affect the full system’s thermodynamics? Prior studies on the stochastic thermodynamics of multipartite processes have addressed this question by assuming that, in addition to the constraints of the dependency network, only one subsystem is allowed to change state at a time. However, in many real systems, such as chemical reaction networks or electronic circuits, multiple subsystems can—or must—change state together. Here, we investigate the thermodynamics of such composite processes, in which multiple subsystems are allowed to change state simultaneously. We first present new, strictly positive lower bounds on entropy production in composite processes. We then present thermodynamic uncertainty relations for information flows in composite processes. We end with strengthened speed limits for composite processes.

## 1. Introduction

Many dynamical systems can be decomposed into a set of multiple co-evolving degrees of freedom. A canonical example is where the separate degrees of freedom are physically distinct variables. As a particularly important example, information-processing systems such as computers and brains consist of many separate components that evolve together and affect each other’s dynamics—these components are the subsystems of the overall system. For brevity, we will refer to the different degrees of freedom co-evolving in an overall system as different **subsystems**.

Often, the dynamics of one of the subsystems will not directly directly depend on the joint state of all the other subsystems, but only on the state of a subset of the other subsystems. Examples range from spin networks to CRNs to biological systems to electronic circuits to full, digital computers. How does the thermodynamics of such systems depend on the network of dependencies in the dynamics of its constituent subsystems?

Thus far, in stochastic thermodynamics, this question has primarily been analyzed by imposing the additional requirement that only one subsystem can change state at any given time. Such systems are known as multipartite processes (MPPs) [1,2,3]. Physically, for a system to be an MPP, it must be that each subsystem is (effectively) coupled to a different set of external reservoirs from the other subsystems.

However, there are many situations where multiple subsystems can—or, in fact, must—change state at the same time. As a canonical example, in the finite size limit of CRNs (Figure 1a), multiple species counts *must* change state concurrently. As another example, the voltages on different conductors in a circuit (Figure 1b) must change state at the same time. Physically, whenever multiple subsystems can change state simultaneously, these subsystems are (effectively) coupled to some of the same external reservoirs. In the language of stochastic thermodynamics, they share “mechanisms”.

Systems with these more general types of coupled subsystems are called **composite systems**, and their overall dynamics is called a **composite process**. There has been some preliminary work extending the stochastic thermodynamics of MPPs to consider composite processes [4]. (It should also be noted that [5] considered scenarios where different subsystems share mechanisms, restricted to the specific issue of how to extend the concept of the “learning rate” to biochemical diffusion processes that have this characteristic.) Here, we extend the preliminary work in [4] and obtain new results on the stochastic thermodynamics of composite processes.

We begin by reviewing some of this earlier work on the stochastic thermodynamics of composite processes. A central feature in the stochastic thermodynamics of a composite process is represented by the”units of the process. As formalized below, these are subsets of all of the subsystems whose joint dynamics does not depend on the state of the rest of the system. (In general, though, the dynamics of *external* subsystems will depend on the dynamics of some of the subsystems within a given unit.) Our first contribution is to illustrate how to apply previous bounds on EP from the literature (e.g., thermodynamic uncertainty relations (TURs [6,7,8,9,10]), kinetic uncertainty relations (KURs [11,12]), mismatch cost bounds on EP [13,14,15,16], speed limit theorems (SLTs [17,18,19]), etc.) to individual units in a composite process, in order to derive strictly positive lower bounds on the EP of the overall system. Crucially, as we show, this can sometimes be done when none of the previous bounds apply to the entire system as a whole, since the conditions for these bounds only apply to individual units, and not to the overall system.

Next, we show how to decompose key quantities of an overall composite process (including probability flows, EP, and dynamical activity) into the contributions of each mechanism. We then combine this decomposition of key quantities with the definition of units to derive a set of new TURs. Finally, we derive a strengthened SLT for composite processes. This speed limit provides a tighter restriction on how much the probability distribution over system states can change during a fixed time interval, using the contributions from each mechanism to EP and dynamical activity. (Note that these results also apply to MPPs, since MPPs are a special case of composite processes.) We conclude by discussing our results in the broader contexts of the thermodynamics of constraints and the thermodynamics of computation and by suggesting avenues for future work.

## 2. Stochastic Thermodynamics of Composite Processes

### 2.1. Background on Composite Processes

A composite process is a generalization of MPPs, describing the Markovian co-evolution of a finite set of subsystems, N={1,2,…,N}. Each subsystem *i* has a discrete state space $X_{i}$ [4]. *x* indicates a state vector in $X=\times_{\;i\in N}X_{i}$, the joint state space of the full system. $x_{A}$ indicates a state vector in $X=\times_{\;i\in A}X_{i}$, the joint state space of the subset *A*. The probability that the entire system is in state *x* at time *t* evolves according to a master equation:(1)$$\begin{array}{c} \frac{d}{dt}p_{x}\left(t\right)=K_{x}^{x^{\prime }}\left(t\right)p_{x^{\prime }}\left(t\right) \end{array}$$Physically, this stochastic dynamics arises due to couplings of the system with a set of mechanisms $V=\{v_{1},v_{2},\ldots ,v_{M}\}$ [20,21]. In general, each such mechanism *v* couples to only a subset of the subsystems. We refer to the set of subsystems to which a mechanism *v* couples as its **puppet set** and write it as P(v)⊆N.

As an example, an MPP is a composite process where each mechanism couples to only one subsystem (although a single subsystem might be coupled to multiple mechanisms [1]). Thus, in an MPP, the cardinality of every puppet set is 1.

At any given time, a composite system changes state due to its interaction with, at most, one mechanism, as with MPPs. Accordingly, the rate matrix of the overall system is a sum of mechanism-specific rate matrices:(2)$$\begin{array}{cc} K_{x}^{x^{\prime }}\left(t\right) & =\sum_{v\in V}\delta_{x_{N\setminus P(v)}}^{x_{N\setminus P(v)}^{\prime }}K_{x_{P(v)},x_{N\setminus P(v)}}^{x_{P(v)}^{\prime },x_{N\setminus P(v)}^{\prime }}\left(t\right) \end{array}$$(3)$$\begin{array}{cc} & :=\sum_{v\in V}K_{x}^{x^{\prime }}\left(v;t\right) \end{array}$$(Here, and throughout, for any two variables $z,z^{\prime }$ contained in the same space, $\delta_{z}^{z^{\prime }}$ is the Kronecker delta function, which equals 1 when $z^{\prime }=z$ and equals 0 otherwise.)

An example of a composite process was presented in [4]. This example involved a particle performing a random walk across a two-dimensional grid. The different subsystems corresponded to different degrees of freedom in a coordinate system for the particle.

As another example, consider a toy stochastic CRN (Figure 1a) [22,23,24]. This network involves four co-evolving species $\left\{X_{1},X_{2},X_{3},X_{4}\right\}$ that change state according to three chemical reactions {A,B,C} (top panel in figure, LHS). The system state is a four-component vector specifying the number of each type of molecule (species) in the system. We identify each of these four separate degrees of freedom as its own subsystem.

Only one reaction can occur at a time, but when a reaction does occur, *multiple* subsystems all change their states. For example, in the forward reaction *A*, species $X_{1}$, $X_{2}$, and $X_{3}$ must change state at the same time, by counts of {−2,−1,+1}, respectively. Accordingly, this reaction network is not an MPP. However, it *is* a composite process.

We can illustrate this composite process in terms of the associated puppet sets. There are a total of three such puppet sets, one for each of the possible chemical reactions. We can denote the mechanisms of the three puppet sets as $r_{A}$, $r_{B}$, and $r_{C}$, with the puppet set of mechanism *r* denoted as P(r). These three puppet sets are indicated by translucent bubbles in the top panel in the figure, RHS.

As a final example, consider a toy electronic circuit [25] consisting of four conductors (the four circles on the left-hand side of Figure 1b) and three devices (the three bidirectional arrows in the figure). The state of the system is a vector consisting of the voltage on each conductor. Two of the conductors (1 and 4) are “regulated”, since they are tied directly to fixed voltage sources ($V_{1}$ and $V_{4}$). The other two conductors (2 and 3) are “free” to stochastically change state via the effect of devices *A*, *B*, and *C*.

The composite process capturing the dynamics of the state of this circuit is illustrated on the right-hand side of the figure. There are three puppet sets (each a translucent bubble), each corresponding to a mechanism associated with one of the devices in the system. The mechanisms are denoted as $r_{A}$, $r_{B}$, and $r_{C}$, and the puppet set of mechanism *r* is denoted as P(r).

In an MPP, although the mechanisms that affect the dynamics of any subsystem *i* do not affect the dynamics of any other subsystem, in general, the dynamics of *i* will depend on the states of some set of other subsystems. For example, in a bipartite process [1,26], both of the subsystems can be modeled as having their own set of mechanisms, but each subsystem’s dynamics is governed by the state of the other subsystem as well as its own state.

Similarly, in a composite process, the dynamics of each subsystem *i* can depend on the states of other subsystems in addition to its own state. Each such dependency can be represented as an edge in a directed graph. In the resulting **dependency network**, each edge j→i means that the state of subsystem *j* affects the rate of state transitions in subsystem *i*. (We do not assign the self-dependency of a subsystem’s dynamics its own edge.) We refer to the set of subsystems whose states affect the dynamics of *i* as the **leaders** of *i*. Thus, j→i means that *j* is a leader of *i*, in addition to *i* itself. In any dependency network, the leaders of each subsystem *i* are *i* itself together with its parents in the dependency graph, pa(i).

The **leader set** for a mechanism *v* is defined to be the union of the leaders of each subsystem in the puppet set of *v*: $ℒ(v)={⋃}_{i\in P(v)}\mathrm{pa}\left(i\right)$, where pa(i) represents the parents of *i* in the dependency network. As an example, although the puppet set of mechanism $v_{2}$ in Figure 2 is {A,C,D}, the leader set of $v_{2}$ is {A,B,C,D}.

The leader set of any mechanism is a (perhaps proper) superset of its puppet set. Accordingly, we can write
(4)$$\begin{array}{c} K_{x}^{x^{\prime }}\left(v;t\right)=K_{x_{ℒ(v)\setminus P(v)}^{\prime },x_{P(v)},x_{N\setminus ℒ(v)}^{\prime }}^{x_{ℒ(v)}^{\prime },x_{N\setminus ℒ(v)}^{\prime }}\left(v;t\right) \end{array}$$With abuse of notation, we can rewrite this in a way that explicitly embodies the fact that the instantaneous dynamics of the puppet set P(v) depends at most on the state of the leader set ℒ(v), and not on the state of any of the subsystems in N∖ℒ(v):(5)$$\begin{array}{c} K_{x_{ℒ(v)\setminus P(v)}^{\prime },x_{P(v)}}^{x_{ℒ(v)}^{\prime }}\left(v;t\right):=K_{x_{P(v)}}^{x_{ℒ(v)}^{\prime }}\left(v;t\right) \end{array}$$

### 2.2. Background on Units

A **unit** ω⊆N is a collection of subsystems such that, as the full system’s state evolves via a master equation according to K(t), the marginal distribution over the states of the unit also evolves according to its own CTMC,
(6)$$\begin{array}{c} \frac{d}{dt}p_{x_{\omega }}\left(t\right)=K_{x_{\omega }}^{x_{\omega }^{\prime }}\left(\omega ;t\right)p_{x_{\omega }^{\prime }}\left(t\right) \end{array}$$
for some associated rate matrix K(ω;t). (See [4] for some additional, technical considerations.) Intuitively, a unit is any set of subsystems whose evolution is independent of the states of the subsystems outside the unit. Typically, a unit is a union of leader sets. In such cases, no subsystem in the unit has parents outside of the unit. Importantly, though, this does not prevent there being a subsystem in the unit that is a leader for some subsystem outside of the unit. Thus, there can be mechanisms whose puppet set contains both a subsystem in a unit and a subsystem outside of the unit. Informally speaking, the boundary of a unit in a dependency network can have outgoing edges, even though it cannot have any incoming edges. However, without loss of generality, we can assume that this is not the case. (See Prop. 2. 1 in [4].)

Any union of units is a unit, and any non-empty intersection of units is a unit [4]. For simplicity, we restrict our attention to systems whose units do not change in time (even though their rate matrices may). Note that the entire system N itself is a unit. We denote the set of all units as $N^{†}$. Thus, the units of a composite process provide a topology over the subsystems in the process, whose open sets are the members of $N^{†}$ together with the empty set. (The dependency network of a composite process should not be confused with the “dependency graph” discussed in [4]. Dependency graphs relate the different units in a composite process to one another, not the different subsystems.) We also write ν(ω) for the set of reservoirs governing the stochastic dynamics of the subsystems in ω, and $K_{x_{\omega }}^{x_{\omega }^{\prime }}\left(v;t\right)$ for the (additive) contribution of reservoir v∈ν(ω) to $K_{x_{\omega }}^{x_{\omega }^{\prime }}\left(\omega ;t\right)$, the rate matrix governing the joint dynamics of the subsystems in ω. (In other words, $K_{x_{\omega }}^{x_{\omega }^{\prime }}\left(v;t\right)$ is a re-expression of the term on the RHS of Equation (7).)

A set of units $N^{*}$ is called a **unit structure** if it obeys the following properties [4].

The union of the units in the set equals N, i.e., they cover N:
$$N^{*}=\left\{\omega_{1},\omega_{2},\ldots \right\}:\underset{\omega \in N^{†}}{⋃}\omega =N$$The set is closed under intersections of its units:

$$\forall \;\left(\omega_{1},\omega_{2}\right)\in {\left(N^{*}\right)}^{2},\;\omega_{1}\cap \omega_{2}\in N^{*}$$From now on, for simplicity, we will restrict our attention to unit structures $N^{*}$ where $N\notin N^{*}$.

Since each separate unit in a unit structure evolves according to its own CTMC, all the usual theorems of stochastic thermodynamics apply to each unit separately, with the expected EP rate of a unit ω at time *t* defined exactly as it is in conventional stochastic thermodynamics [20,21]:(7)$$\begin{array}{cc} {\dot{\sigma }}^{\omega }\left(t\right) & =\sum_{x_{\omega }^{\prime },x_{\omega },v\in \nu \left(\omega \right)}K_{x_{\omega }}^{x_{\omega }^{\prime }}\left(v;t\right)p_{x_{\omega }^{\prime }}\left(t\right)\ln\left[\frac{K_{x_{\omega }}^{x_{\omega }^{\prime }}\left(v;t\right)p_{x_{\omega }^{\prime }}\left(t\right)}{K_{x_{\omega }^{\prime }}^{x_{\omega }}\left(v;t\right)p_{x_{\omega }}\left(t\right)}\right] \end{array}$$In particular, the Second Law applies to any unit, as do the mismatch cost theorems. Similarly, when their conditions hold for a particular unit, the associated TURs hold for this unit, and the same is true for the SLTs, the fluctuation theorems [27], and thermodynamic bounds based on first-passage times and stopping times [28,29,30].

In addition to these previous results of stochastic thermodynamics, the relationships among the units in a unit structure provide some new stochastic thermodynamics theorems as well. As an example, for any pair of nested units ω and α⊆ω,
(8)$$\begin{array}{c} {\dot{\sigma }}^{\omega }\left(t\right)\geq {\dot{\sigma }}^{\alpha }\left(t\right) \end{array}$$Loosely speaking, this can be viewed as an extension of the well-known result that ignoring the degrees of freedom in a system cannot increase the associated EP, only decrease it [20]. (See [3,4] for the derivation and discussion of this and related results.) In the sequel, we follow conventional notation, dropping the dot over σ to indicate the integrated value of the EP rate across some time interval.

Let $N^{*}=\left\{\omega_{j}:j=1,2,\ldots ,n\right\}$ be a unit structure. In particular, for all $i,j\in \left\{1,2,\ldots ,n\right\},\omega_{j}\cap \omega_{i}=\omega_{k}$ for some $\omega_{k}\in N^{*}$. Suppose that we have a set of real numbers, *f*, which are indexed (using superscripts) by the units in $N^{*}$. It will be convenient to use the associated shorthand,
(9)$$\begin{array}{cc} {\hat{\sum }}_{\omega \in N^{*}}f^{\omega } & :=\sum_{j=1}^{n}f^{\omega_{j}}-\sum_{1\leq j<j^{\prime }\leq n}f^{\omega_{j}\cap \omega_{j^{\prime }}}+\sum_{1\leq j<j^{\prime }<j^{\prime \prime }\leq n}f^{\omega_{j}\cap \omega_{j^{\prime }}\cap \omega_{j^{\prime \prime }}}-\ldots \end{array}$$
which is well defined because the set of units in a unit structure is closed under intersection. (Note that the precise assignment of integer indices to the units in $N^{*}$ does not affect the value of this sum.) This quantity is called the **inclusion–exclusion sum** (or simply “in–ex sum” for short) of *f* for the unit structure $N^{*}$. (See [31] for background on the inclusion–exclusion principle.)

Next, define the time-*t* **in–ex information** as
(10)IN*:=∑ω∈N*^Sω−SN=−SN+∑j=1nSωj−∑1≤j<j′≤nSωj∩ωj′+…
where all the terms in the sums on the RHS are marginal entropies over the (distributions over the coordinates in the) indicated units. As an example, if $N^{*}$ consists of two units, $\omega_{1},\omega_{2}$, with no intersection, then the expected in–ex information at time *t* is simply the mutual information between these units at that time. More generally, if there is an arbitrary number of units in $N^{*}$ but none of them overlap, then the expected in–ex information is what is called the “multi-information”, or “total correlation”, among these units [32].

In App. E in [4], it is proven that the global EP incurred during a time period [0,τ] can be decomposed as
(11)$$\begin{array}{c} \sigma^{N}={\hat{\sum }}_{\omega \in N^{*}}\sigma^{\omega }-\Delta I^{N^{*}} \end{array}$$Intuitively, the negative-valued terms in the in–ex sums correct for the overcounting of the thermodynamic quantities arising due to the fact that units can overlap with one another. (The formal proof of Equation (11) is somewhat elaborate, using Rota’s extension of the inclusion–exclusion principle to negative-valued measures and the fact—proven in App. M in [4]— that the heat flow into the unit structure decomposes into an in–ex sum.)

## 3. Strictly Positive Lower Bounds on EP from Its In–Ex Decomposition

In some situations, some of the many thermodynamic bounds (TURs, SLTs, mismatch cost, KURs, etc.) will apply separately to some of the units within a system, but no such bound will apply to the overall system. The in–ex sum decomposition of EP can sometimes be used to “knit together” those bounds on the EP generated by the individual units, to give a bound on the EP generated by the overall system.

### 3.1. Mismatch Cost

We can illustrate this by exploiting a recent result concerning the minimal EP generated in any dynamic process [13,14,15,16,33]. Suppose that we have a (perhaps stochastic) dynamic process, taking any initial distribution p(x) to an ending distribution $p^{\prime }\left(x\right)=\sum_{x^{\prime }}P\left(x|x^{\prime }\right)p\left(x\right)$, which we express in shorthand as $p^{\prime }=Pp$. (Thus, *P* is the conditional distribution of the ending state given the initial state.) Write σ(p) for the EP generated by running this process on the distribution *p*. Let *q* be the initial distribution minimizing σ(q) for this fixed process. Then, it is always (exactly) true that for any initial distribution *p*,
(12)$$\begin{array}{cc} \sigma (p)-\sigma (q) & =D(p\;||\;q)-D(Pp\;||\;Pq)\geq 0 \end{array}$$
where D(.||.) is the relative entropy (Kullback–Leibler divergence).

In the literature, *q* is called the **prior**, due to its Bayes optimality when running a full thermodynamic cycle [33], and the drop in KL divergence in Equation (12) is called the **mismatch cost**. (It is important to note that Equation (12) applies for an extremely wide range of dynamic processes implementing *P*, for many types of state space, and for many other thermodynamic costs generated during the process, in addition to EP; see [15,34].)

Now, suppose that there are multiple possible initial distributions, indexed by a (countable) variable θ, $\left\{p_{\theta }\right\}$. Suppose as well that θ∼R for some distribution *R*. Thus, the expected mismatch cost for the fixed process *P* is
(13)$$\begin{array}{c} E_{R}\left[D(p_{\theta }\;||\;q)-D(Pp_{\theta }\;||\;Pq)\right] \end{array}$$In contrast, the mismatch cost for the expected initial distribution is
(14)$$\begin{array}{c} D(E_{R}\left(p_{\theta }\right)\;||\;q)-D(PE_{R}\left(p_{\theta }\right)\;||\;Pq) \end{array}$$The difference between these two quantities is
(15)$$\begin{array}{c} \left[E_{R}\left(S\left(p_{\theta }\right)\right)-S\left(E_{R}\left(p_{\theta }\right)\right)\right]-\left[E_{R}\left(S\left(Pp_{\theta }\right)\right)-S\left(PE_{R}\left(p_{\theta }\right)\right)\right] \end{array}$$
where S(.) is Shannon entropy as usual. This difference equals $\Delta JS_{R(\theta )}\left(\left\{p\left(\theta \right)\right\}\right))$, the change during the process *P* in the value of the Jensen–Shannon (JS) divergence for the set of distributions $p_{\theta }$ where θ is distributed according to R(θ) [35].

Note that $\Delta JS_{R(\theta )}\left(\left\{p\left(\theta \right)\right\}\right))$ is independent of *q*. Accordingly, if we subtract and add the expression in Equation (14) from the expression in Equation (13), and then minimize over priors *q*, we obtain
(16)$$\begin{array}{cc} & \;\;\;\;\;\min_{q}\left(E_{R}[D(p_{\theta }\;||\;q)-D(Pp_{\theta }\;||\;Pq)\right])\\ & \;=\Delta JS_{R(\theta )}\left(\left\{p\left(\theta \right)\right\}\right))+\min_{q}\left(D(E_{R}\left(p_{\theta }\right)\;||\;q)-D(PE_{R}\left(p_{\theta }\right)\;||\;Pq)\right)\\ & \;=\Delta JS_{R(\theta )}\left(\left\{p\left(\theta \right)\right\}\right)) \end{array}$$
where the last equation uses the fact that the expression in Equation (14) is minimized (to 0) by taking $q=E_{R}p_{\theta }$. (A variant of this result was first derived in [33], where JS divergence was called “entropic variance”. See also [36,37].)

### 3.2. Periodic Processes

Next, suppose that we have a process over a space *X* that is periodic, in the sense that, for some real number λ>0,
(17)$$\begin{array}{c} P(x(n\lambda )\;|\;x((n-1)\lambda ))=P(x(\lambda )\;|\;x(0)) \end{array}$$
is the same for all integers n>1. Then, the EP generated during *N* periods starting at t=0, σ(Nλ), is lower-bounded by summing the mismatch cost over all *N* periods [34,37]:(18)$$\begin{array}{c} \sigma \left(N\lambda \right)\geq \underset{q\in \Delta_{X}}{\inf}\sum_{t=0}^{N-1}\left[D(P^{t}p_{0}\;||\;q)-D(P^{t+1}p_{0}\;||\;Pq)\right] \end{array}$$
where $P^{t}$ is the conditional distribution P(x(λ)|x(0)) iterated *t* times, and $\Delta_{X}$ is the unit simplex over *X*.

Each term in the sum on the RHS of Equation (18) is non-negative, by the data-processing inequality of KL divergence. Moreover, so long as the conditional distribution P(x(λ)|x(0)) is non-degenerate (i.e., not simply a deterministic permutation over the states of *X*) and the initial distribution $p_{0}$ is not the steady state of *P*, at most, one of the terms in this sum on the RHS of Equation (18) can equal zero; all the other terms must be strictly positive. In such a situation, Equation (18) provides a strictly positive lower bound on the EP, σ(Nλ).

Indeed, we can re-express the sum on the RHS of Equation (18) as the uniform average over a set of *t*-parameterized distributions, $\left\{P^{t}p_{0}\right\}$, of the mismatch cost between this distribution and *q*. Inserting it into Equation (16), this establishes that
(19)$$\begin{array}{c} \sigma \left(N\lambda \right)\geq N\Delta JS_{U_{N}\left(t\right)}\left(\left\{P^{t}p_{0}\right\}\right) \end{array}$$
where $U_{t}$ is the uniform distribution over the values t=1,…,N.

This lower bound in EP in Equation (19) is defined entirely in terms of the conditional distribution *P* and the initial distribution $p_{0}$. None of the particular physical details of how the periodic process is physically implemented are involved in this “periodicity mismatch cost” lower bound on EP. In the sense of this generality, it is similar to the generalized Landauer’s bound. In particular, the TURs and SLTs are also lower bounds on EP that depend on the initial distribution over states and the discrete time conditional distribution of the dynamics. However, unlike the periodicity mismatch cost lower bound, the other bounds depend on other properties of the process, besides the initial distribution and the conditional distribution giving the dynamics. (For example, they depend on factors such as current precision or expected activities.) In this sense, the periodicity mismatch cost bound is more powerful than the other lower bounds on EP.

### 3.3. Example Where In–Ex Decomposition Is Necessary for Lower-Bound EP

In this subsection, we show how to insert Equation (18) into the in–ex decomposition of EP, Equation (11), and thereby derive a strictly positive lower bound on the EP of the full system—a lower bound that we could not derive without using this decomposition.

Suppose that we have three subsystems, A,B,C, of an overall system co-evolving with state spaces $X_{A},X_{B}$, and $X_{C}$, accordingly. Further, suppose that A,AC, and AB are the units in a unit structure. Suppose as well that *A* is in a meta-stable state (e.g., due to high energy barriers) throughout the co-evolution.

Consider the case where *B* undergoes an exactly periodic process parameterized by $x_{A}$, i.e., given that $x_{A}$ does not change in time, there is some time interval $\lambda_{B}$ such that
(20)$$\begin{array}{cc} & P\left(x_{B}\left(n\lambda_{B}\right)\;|\;x_{B}\left(\left(n-1\right)\lambda_{B}\right),x_{A}\left(0\right)\right)=P\left(x_{B}\left(\lambda_{B}\right)\;|\;x_{B}\left(0\right)\right) \end{array}$$
is the same conditional distribution for all integers n>1. Thus, the dynamics of *B* is a time-homogeneous discrete-time Markov chain over timesteps given by the value of *n*, with the *n*-independent Markov kernel $P_{AB}\left(x_{B}\left(n+1\right)\;|\;x_{B}\left(n\right),x_{A}\left(0\right)\right)$. Similarly, $P(x_{C}\left(n\lambda_{C}\right)\;|\;x_{C}\left(\left(n-1\right)\lambda_{C}\right),x_{A}\left(0\right))$ is the same conditional distribution for all integers n>1. Thus, the dynamics of *C* is a time-homogeneous discrete-time Markov chain over timesteps given by the value of *n*, with an *n*-independent Markov kernel, namely $P_{AC}\left(x_{C}\left(n+1\right)\;|\;x_{C}\left(n\right),x_{A}\left(0\right)\right)$.

Choose $\lambda_{B}$ and $\lambda_{C}$ so that there are two coprime positive integers m,n>m such that $\lambda_{B}=m\lambda_{C}/n$, and m>2. Thus, whatever the value of $x_{A}$, the dynamics of the overall system is periodic with period $\tau :=n\lambda_{B}=m\lambda_{C}$.

Suppose that we are interested in the EP generated by the overall system in the time interval [0,τ]. The RHS of Equation (18) is zero for the overall system for this time interval, since τ is merely a single period τ of the overall system, and so the infimum equals zero regardless of the dynamics or the initial distribution (simply take $q=p_{0}$). Thus, Equation (18) cannot be used directly to bound the EP of the overall system.

However, we can apply Equation (11) to the overall system and then use Equation (18) to lower-bound the EP generated by the units AB and AC. (Note that since *A* is in a meta-stable state, it generates no EP.)

To do this, first extend the definition given below Equation (10) to write the conditional entropy of the variables in a unit α conditioned on the variables in a unit γ as $S^{\alpha |\gamma }$. With this notation, by exploiting the fact that the entropy over the states of *A* does not change during the process, we can expand Equation (10) for our unit structure as
(21)$$\begin{array}{cc} -\Delta I^{N^{*}} & \;:=\;\Delta S^{ABC}-\Delta S^{AC}-\Delta S^{AB} \end{array}$$
(22)$$\begin{array}{cc} & =\Delta S^{BC|A}-\Delta S^{AC|A}-\Delta S^{AB|A} \end{array}$$
(23)$$\begin{array}{cc} & =-\Delta I(X_{B};X_{C}\;|\;X_{A}) \end{array}$$
where the Δ*S* indicate the changes in a quantity from t=0 to t=τ, and $I(X_{B};X_{C}\;|\;X_{A})$ is the usual conditional mutual information.

Next, to evaluate $\Delta I(X_{B};X_{C}\;|\;X_{A})$, note that since $x_{A}\left(t\right)$ is independent of time, the dynamics of $x_{B}\left(t\right)$ is a function of $x_{B}\left(t\right)$ and $x_{A}\left(0\right)$, conditionally independent of the state $x_{C}\left(t\right)$; similarly, the dynamics of $x_{C}\left(t\right)$ is conditionally independent of the state $x_{B}\left(t\right)$. In other words, for each value $x_{A}\left(0\right)\in X_{A}$,
(24)$$\begin{array}{cc} & \left(x_{B}\left(\tau \right),x_{A}\left(0\right)\right)\to \left(x_{B}\left(0\right),x_{A}\left(0\right)\right)\to (x_{C}\left(0\right),x_{A}\left(0\right)\to \left(x_{C}\left(\tau \right),x_{A}\left(0\right)\right) \end{array}$$
is a Markov chain. Therefore, by the data-processing inequality [38], for each value of $x_{A}\left(0\right)$, the mutual information between $\left(x_{B}\left(\tau \right),x_{A}\left(0\right)\right)$ and $\left(x_{C}\left(\tau \right),x_{A}\left(0\right)\right)$ is not greater than the mutual information between $\left(x_{B}\left(0\right),x_{A}\left(0\right)\right)$ and $\left(x_{C}\left(0\right),x_{A}\left(0\right)\right)$. This establishes that $-\Delta I(X_{B};X_{C}\;|\;X_{A})\geq 0$, and so, by Equation (23),
(25)$$\begin{array}{c} -\Delta I^{N^{*}}\geq 0 \end{array}$$

Inserting Equation (23) into Equation (11) and then using Equation (19) twice, once for the unit AC and once for the unit AB, we can lower-bound the EP generated by the full system in the time interval [0,τ]:(26)$$\begin{array}{cc} \sigma^{N} & ={\hat{\sum }}_{\omega \in N^{*}}\sigma^{\omega }-\Delta I^{N^{*}} \end{array}$$(27)$$\begin{array}{cc} & =\sigma^{AB}+\sigma^{AC}-\Delta I^{N^{*}}\\ & \geq -\Delta I^{N^{*}}+\underset{q\in \Delta_{X_{A}\times X_{B}}}{\inf}\sum_{t=0}^{n-1}\left[D(P_{AB}^{t}p_{0}\left(X_{A},X_{B}\right)\;||\;q\left(X_{A},X_{B}\right))-D(P_{AB}^{t+1}p_{0}\left(X_{A},X_{B}\right)\;||\;P_{AB}q\left(X_{A},X_{B}\right))\right] \end{array}$$(28)$$\begin{array}{cc} & \;+\underset{q\in \Delta_{X_{A}\times X_{C}}}{\inf}\sum_{t=0}^{m-1}\left[D(P_{AC}^{t}p_{0}\left(X_{A},X_{C}\right)\;||\;q\left(X_{A},X_{C}\right))-D(P_{AB}^{t+1}p_{0}\left(X_{A},X_{C}\right)\;||\;P_{AC}q\left(X_{A},X_{C}\right))\right] \end{array}$$(29)$$\begin{array}{cc} & =-\Delta I^{N^{*}}+\Delta JS_{U_{n}\left(t\right)}\left(\left\{P_{AB}^{t}p_{0}\left(X_{A},X_{B}\right)\right\}\right)+\Delta JS_{U_{m}\left(t\right)}\left(\left\{P_{AC}^{t}p_{0}\left(X_{A},X_{C}\right)\right\}\right) \end{array}$$

Every term in this lower bound on the EP of the full system is non-negative (including all the terms in the two sums), and if $p_{0}$ is not a steady state of the dynamics, while both $P_{AB}$ and $P_{BC}$ are non-degenerate, at most one term in each of the two sums can be zero. In such a case, Equation (29) provides a strictly positive lower bound on the EP of the overall system. Moreover, this bound can be evaluated knowing only the values of $\lambda_{B},\lambda_{C},\tau ,p_{0}\left(x_{A},x_{B},x_{C}\right)$ and the two conditional distributions $P_{AB}$ and $P_{AC}$, giving the periodic dynamics of units AB and AC, respectively. All other details of the physical process implementing the dynamics are irrelevant.

Ultimately this bound depends on the insertion of Equation (19) into the in–ex decomposition, Equation (11). In contrast, as mentioned above, we cannot establish a strictly positive lower bound on the EP of the full system by using Equation (19) only, without also using the in–ex decomposition.

In this example, a single lower bound on EP from the literature was combined with the in–ex decomposition to provide a strictly positive lower bound on EP, where no such bound could be derived without using this decomposition. This same method of exploiting the in–ex decomposition of EP can also be used to “mix and match” different thermodynamic bounds (TURs, SLTs, mismatch costs, KURs, etc.), each of which applies to different units, knitting those bound together to give a bound on the EP of the overall system. See [2] for some examples for the special case of MPPs.

## 4. Thermodynamics Due to Multiplicity of Mechanisms

None of the results presented thus far rely on the overall system obeying local detailed balance. In fact, they do not depend on there being a Hamiltonian function for the overall system. In the remainder of this paper, we will implicitly restrict our attention to systems that *do* obey local detailed balance, so that our results have direct thermodynamic significance. In addition, since the entire system is itself a unit, we will write all our results in terms of arbitrary units ω for the rest of the paper; they apply even if all of N is the *only* unit in our system.

### 4.1. Additional Decompositions of Thermodynamic and Dynamical Quantities in Composite Processes

The rate matrix of each unit ω in a composite process decomposes into rate matrices from each mechanism whose leader set is a subset of ω:(30)$$\begin{array}{cc} K_{x_{\omega }}^{x_{\omega }^{\prime }}\left(\omega ;t\right) & =\sum_{v:ℒ(v)\subseteq \omega }\delta_{x_{\omega \setminus ℒ(v)}}^{x_{\omega \setminus ℒ(v)}^{\prime }}K_{x_{ℒ(v)},x_{\omega \setminus ℒ(v)}}^{x_{ℒ(v)}^{\prime },x_{\omega \setminus ℒ(v)}^{\prime }}\left(t\right) \end{array}$$(31)$$\begin{array}{cc} & =\sum_{v:ℒ(v)\subseteq \omega }K_{x_{\omega }}^{x_{\omega }^{\prime }}\left(v;t\right) \end{array}$$Similarly, we can decompose the EP rate of any unit ω into contributions ${\dot{\zeta }}_{\omega }^{v}\left(t\right)$ from each mechanism whose leader set is a subset of ω:(32)$$\begin{array}{cc} {\dot{\sigma }}^{\omega }\left(t\right) & =\sum_{\begin{array}{c} v:ℒ(v)\subseteq \omega ,\\ x_{\omega }^{\prime },x_{\omega }\neq x_{\omega }^{\prime } \end{array}}K_{x_{\omega }}^{x_{\omega }^{\prime }}\left(v;t\right)\;p_{x_{\omega }^{\prime }}\left(t\right)\ln\left[\frac{K_{x_{\omega }}^{x_{\omega }^{\prime }}\left(v;t\right)\;p_{x_{\omega }^{\prime }}\left(t\right)}{K_{x_{\omega }^{\prime }}^{x_{\omega }}\left(v;t\right)\;p_{x_{\omega }}\left(t\right)}\right] \end{array}$$(33)$$\begin{array}{cc} & =\sum_{v:ℒ(v)\subseteq \omega }{\dot{\zeta }}_{\omega }^{v}\left(t\right) \end{array}$$In particular, since the entire system is a unit whose state transitions are mediated by every mechanism v∈V, the global EP rate decomposes as ${\dot{\sigma }}^{N}\left(t\right)=\sum_{v}{\dot{\zeta }}_{N}^{v}\left(t\right)$.

A unit’s dynamical activity also decomposes:(34)$$\begin{array}{c} A^{\omega }\left(t\right)=\sum_{\begin{array}{c} v:ℒ(v)\subseteq \omega ,\\ x_{\omega }^{\prime },x_{\omega }\neq x_{\omega }^{\prime } \end{array}}K_{x_{\omega }}^{x_{\omega }^{\prime }}\left(v;t\right)\;p_{x_{\omega }^{\prime }}\left(t\right)=\sum_{v:ℒ(v)\subseteq \omega }A\left(v;t\right) \end{array}$$Similarly, the entire system’s dynamical activity can be decomposed as $A^{N}\left(t\right)=\sum_{v}A\left(v;t\right)$. Note that the dynamics of every pair of nested units ω,α⊆ω must be consistent [4], which means that $A_{\alpha }\left(v;t\right)=A_{\omega }\left(v;t\right)=A\left(v;t\right)$ for all α and ω.

We denote the probability flow from $x_{\omega }^{\prime }\to x_{\omega }$ due to mechanism *v* as $A_{x_{\omega }}^{x_{\omega }^{\prime }}\left(v;t\right)=K_{x_{\omega }}^{x_{\omega }^{\prime }}\left(v;t\right)\;p_{x_{\omega }^{\prime }}\left(t\right)$. We write the *net* probability current from $x_{\omega }^{\prime }\to x_{\omega }$ due to mechanism *v* as $J_{x_{\omega }}^{x_{\omega }^{\prime }}\left(v;t\right)=A_{x_{\omega }}^{x_{\omega }^{\prime }}\left(v;t\right)-A_{x_{\omega }^{\prime }}^{x_{\omega }}\left(v;t\right)$. (Note that this is not defined for $x_{\omega }^{\prime }=x_{\omega }$.) The total net probability current from $x_{\omega }^{\prime }\to x_{\omega }$ equals the sum of the probability currents due to each mechanism whose leader set is a subset of the unit ω:(35)$$\begin{array}{c} J_{x_{\omega }}^{x_{\omega }^{\prime }}\left(t\right)=\sum_{v:ℒ(v)\subseteq \omega }J_{x_{\omega }}^{x_{\omega }^{\prime }}\left(v;t\right) \end{array}$$Accordingly, we can decompose the master equation for the unit ω into the probability currents induced by each mechanism:(36)$$\begin{array}{c} \frac{d}{dt}p_{x_{\omega }}\left(t\right)=\sum_{v:ℒ(v)\subseteq \omega }K_{x_{\omega }}^{x_{\omega }^{\prime }}\left(v;t\right)p_{x_{\omega }^{\prime }}\left(t\right)=\sum_{\begin{array}{c} v:ℒ(v)\subseteq \omega ,\\ x_{\omega }^{\prime }\neq x_{\omega } \end{array}}J_{x_{\omega }}^{x_{\omega }^{\prime }}\left(v;t\right) \end{array}$$

### 4.2. Thermodynamic Uncertainty Relations for Composite Processes

For any unit ω that is in an NESS, any linear function of probability currents ${𝒞}_{\omega }$ is a current. It can be divided into the contributions of each mechanism:(37)$$\begin{array}{cc} {\dot{𝒞}}_{\omega } & =\sum_{x_{\omega }^{\prime },x_{\omega }>x_{\omega }^{\prime }}J_{x_{\omega }}^{x_{\omega }^{\prime }}C_{x_{\omega }}^{x_{\omega }^{\prime }} \end{array}$$(38)$$\begin{array}{cc} & =\sum_{\begin{array}{c} v:ℒ(v)\subseteq \omega ,\\ x_{\omega }^{\prime },x_{\omega }>x_{\omega }^{\prime } \end{array}}J_{x_{\omega }}^{x_{\omega }^{\prime }}\left(v\right)C_{x_{\omega }}^{x_{\omega }^{\prime }} \end{array}$$(39)$$\begin{array}{cc} & =\sum_{v:ℒ(v)\subseteq \omega }{\dot{𝒞}}_{\omega }\left(v\right) \end{array}$$
where $C_{x_{\omega }}^{x_{\omega }^{\prime }}=-C_{x_{\omega }^{\prime }}^{x_{\omega }}$ is some anti-symmetric function of state transitions, and we have dropped the time dependence in the steady state.

Importantly, the current contribution from each mechanism ${\dot{𝒞}}_{\omega }\left(v\right)$ is itself a current. Thus, all of the thermodynamic uncertainty relations (TURs) hold for the time-integrated version of any such mechanism-specific current. In an NESS running for a time period of length τ, this mechanism-specific time-integrated current is ${𝒞}_{\omega }\left(v\right)=\tau {\dot{𝒞}}_{\omega }\left(v\right)$. Additionally, since every unit evolves according to its own CTMC, the TURs hold for each unit.

For example, the finite-time TUR bounds the precision of any current in a CTMC with respect to its EP rate [8,39]. For a composite process, this holds for any unit and any arbitrary time-integrated current:(40)$$\begin{array}{c} \sigma^{\omega }\geq \frac{2{\langle {𝒞}_{\omega }\rangle }^{2}}{\mathrm{Var}({𝒞}_{\omega })} \end{array}$$Additionally, for any mechanism v:ℒ(v)⊆ω and any associated current ${𝒞}_{\omega }\left(v\right)$,
(41)$$\begin{array}{c} \sigma^{\omega }\geq \frac{2{\langle {𝒞}_{\omega }\left(v\right)\rangle }^{2}}{\mathrm{Var}({𝒞}_{\omega }\left(v\right))} \end{array}$$

The vector-valued TUR following [9] holds for a vector ${\dot{𝒞}}_{\omega }$ of any set of (potentially mechanism-specific) currents $\left\{{\dot{𝒞}}_{\omega }\right\}$ that are not linearly dependent:(42)$${\dot{𝒞}}_{\omega }^{T}\;\Xi_{\omega }^{-1}\;{\dot{𝒞}}_{\omega }\leq \frac{{\dot{\sigma }}^{\omega }}{2\tau }$$
where $\Xi_{\omega }^{-1}$ is the inverse of the covariance matrix of the associated time-integrated currents $\left\{{𝒞}_{\omega }\right\}$.

Any of these TURs can be useful to bound the EP when one has limited access to the system in the sense that one can measure state transitions (i) due only to some subset of the mechanisms influencing the system or state transitions or (ii) involving some subset of units in the system.

### 4.3. Information Flow TURs

One important quantity in an MPP is information flow [1,26,40]. Here, we extend the concept of information flow to composite processes. For any unit ω in an NESS, a set of subsystems A⊂ω, and a set of subsystems B⊂ω (for which A∩B=⌀), the information flow is the rate of decrease in the conditional entropy of the state of *B* given the state of *A*, due to state transitions in *A*:(43)$$\begin{array}{c} {\dot{I}}^{A\to B}=\sum_{x_{\omega }^{\prime },x_{\omega }>x_{\omega }^{\prime }}J_{x_{\omega }}^{x_{\omega }^{\prime }}\delta_{x_{\omega \setminus A}}^{x_{\omega \setminus A}^{\prime }}\ln\frac{p_{x_{B}|x_{A}}}{p_{x_{B}|x_{A}^{\prime }}} \end{array}$$Thus, when ω is in an NESS, the information flow is a current for which $C_{x_{\omega }}^{x_{\omega }^{\prime }}=C_{x_{\omega \setminus A},x_{A}}^{x_{\omega \setminus A}^{\prime },x_{A}^{\prime }}=\delta_{x_{\omega \setminus A}}^{x_{\omega \setminus A}^{\prime }}\ln\frac{p_{x_{B}|x_{A}}}{p_{x_{B}|x_{A}^{\prime }}}$. The contribution to the information flow that is due to interactions of the unit with reservoir v:ℒ(v)⊆ω is itself an information flow:(44)$$\begin{array}{c} {\dot{I}}^{A\to B}\left(v\right)=\sum_{x_{\omega }^{\prime },x_{\omega }>x_{\omega }^{\prime }}J_{x_{\omega }}^{x_{\omega }^{\prime }}\left(v\right)\delta_{x_{\omega \setminus A}}^{x_{\omega \setminus A}^{\prime }}\ln\frac{p_{x_{B}|x_{A}}}{p_{x_{B}|x_{A}^{\prime }}} \end{array}$$Since these information flows are currents, the TURs will apply to them. This observation, in combination with Equation (8), suggests that the precision of an information flow is (best) bounded by the reciprocal of the EP of the smallest unit that contains A∪B.

## 5. Strengthened Thermodynamic Speed Limits for Composite Processes

Here, we derive a speed limit similar to the one in [19], but for composite processes. This speed limit is tighter than the one presented in the mentioned paper. Our analysis will hold for an arbitrary unit ω (which could be the entire system N itself),
(45)$$\begin{array}{c} l_{\omega }\leq \sum_{v:ℒ(v)\subseteq \omega }A_{\omega }^{\mathrm{tot}}\left(v;\tau \right)\;f\left(\frac{\zeta_{\omega }^{v}\left(\tau \right)}{A_{\omega }^{\mathrm{tot}}\left(v;\tau \right)}\right) \end{array}$$
where the dynamics occurs during the time period [0,τ]. Additionally, $l_{\omega }$ is the total variation distance between the initial (time-0) and final (time-τ) probability distributions over states of the unit ω. $A_{\omega }^{\mathrm{tot}}\left(v;\tau \right)$ is the total time-integrated dynamical activity due to mechanism *v*. $\zeta_{\omega }^{v}\left(\tau \right)$ is the total contribution to the EP of unit ω due to interactions of ω with mechanism *v*.

We start by bounding the total variation distance between the initial and final (time-τ) probability distributions over states of the unit ω:(46)$$\begin{array}{cc} l_{\omega } & :=𝕃(p_{x_{\omega }}\left(0\right),p_{x_{\omega }}\left(\tau \right))=\frac{1}{2}\sum_{x_{\omega }}|p_{x_{\omega }}\left(\tau \right)-p_{x_{\omega }}\left(0\right)| \end{array}$$(47)$$\begin{array}{cc} & =\frac{1}{2}\sum_{x_{\omega }}\left|\int_{0}^{\tau }dt\frac{d}{dt}p_{x_{\omega }}\left(t\right)\right| \end{array}$$(48)$$\begin{array}{cc} & \leq \frac{1}{2}\int_{0}^{\tau }dt\sum_{x_{\omega }}\left|\frac{d}{dt}p_{x_{\omega }}\left(t\right)\right| \end{array}$$In a composite process, we can further bound the integrand:(49)$$\begin{array}{cc} \sum_{x_{\omega }}|\frac{d}{dt}p_{x_{\omega }}\left(t\right)| & =\sum_{x_{\omega }}|\sum_{v:ℒ(v)\subseteq \omega }\sum_{x_{\omega }^{\prime }\neq x_{\omega }}J_{x_{\omega }}^{x_{\omega }^{\prime }}\left(v;t\right)| \end{array}$$(50)$$\begin{array}{cc} & \leq \sum_{v:ℒ(v)\subseteq \omega }\sum_{x_{\omega },x_{\omega }^{\prime }\neq x_{\omega }}\left|J_{x_{\omega }}^{x_{\omega }^{\prime }}\left(v;t\right)\right| \end{array}$$We write the time-*t* “conditional probability distribution” of the forward process, under the counterfactual scenario whereby the process evolves with coupling only to mechanism v:ℒ(v)⊆ω, as
(51)$$\begin{array}{c} W_{x_{\omega }}^{x_{\omega }^{\prime }}\left(v;t\right)=\frac{\left(1-\delta_{x_{\omega }}^{x_{\omega }^{\prime }}\right)\;K_{x_{\omega }}^{x_{\omega }^{\prime }}\left(v;t\right)\;p_{x_{\omega }^{\prime }}\left(t\right)}{A_{\omega }\left(v;t\right)} \end{array}$$Intuitively, this can be interpreted as a conditional probability that if a jump occurs at *t* due to reservoir v:ℒ(v)⊆ω, the state before the jump was $x_{\omega }^{\prime }$ and the state afterwards was $x_{\omega }$. We write the same quantity for the reverse process as
(52)$$\begin{array}{c} {\tilde{W}}_{x_{\omega }}^{x_{\omega }^{\prime }}\left(t\right)=\frac{\left(1-\delta_{x_{\omega }^{\prime }}^{x_{\omega }}\right)\;K_{x_{\omega }^{\prime }}^{x_{\omega }}\left(v;t\right)\;p_{x_{\omega }}\left(t\right)}{A_{\omega }\left(v;t\right)} \end{array}$$The total variation distance between these matrices $d_{\mathrm{TV}}\left(W_{\omega }\left(v;t\right),{\tilde{W}}_{\omega }\left(v;t\right)\right)$ represents how irreversible this counterfactual process (the one driven only by mechanism *v*) is at time *t*. Using these definitions, we can rewrite Equation (50) as
(53)$$\begin{array}{cc} \sum_{x_{\omega }} & \left|\frac{d}{dt}p_{x_{\omega }}\left(t\right)\right|\leq 2\sum_{v:ℒ(v)\subseteq \omega }A_{\omega }\left(v;t\right)\;d_{\mathrm{TV}}\left(W_{\omega }\left(v;t\right),{\tilde{W}}_{\omega }\left(v;t\right)\right) \end{array}$$Inserting this into Equation (48), we obtain
(54)$$\begin{array}{c} l_{\omega }\leq \int_{0}^{\tau }dt\sum_{v:ℒ(v)\subseteq \omega }A_{\omega }\left(v;t\right)\;d_{\mathrm{TV}}\left(W_{\omega }\left(v;t\right),{\tilde{W}}_{\omega }\left(v;t\right)\right) \end{array}$$

We next make use of the fact that mechanism *v*’s contribution to the EP rate of unit ω (Equation (33)) can be written in terms of the Kullback–Leibler (KL) divergence between the conditional distributions of the forward and backward processes as
(55)$$\begin{array}{c} {\dot{\zeta }}_{\omega }^{v}\left(t\right)=A_{\omega }\left(v;t\right)\;D_{\mathrm{KL}}\left(W_{\omega }\left(v;t\right),{\tilde{W}}_{\omega }\left(v;t\right)\right) \end{array}$$Some positive monotonic concave functions g(.) relate the total variation distance to the KL divergence [19] according to
(56)$$\begin{array}{c} d_{\mathrm{TV}}\left(p;q\right)\leq g\left(D_{\mathrm{KL}}\left(p;q\right)\right) \end{array}$$We can use this relationship to relate Equation (55) to $l_{\omega }$. Combining Equations (54)–(56),
(57)$$\begin{array}{cc} l_{\omega } & \leq \int_{0}^{\tau }dt\sum_{v:ℒ(v)\subseteq \omega }A_{\omega }\left(v;t\right)\;f\left(\frac{{\dot{\zeta }}_{\omega }^{v}\left(t\right)}{A_{\omega }\left(v;t\right)}\right) \end{array}$$

Next, define $\zeta_{\omega }^{v}=\int_{0}^{\tau }dt{\dot{\zeta }}_{\omega }^{v}\left(t\right)$ as the total (ensemble-average) contribution to the EP of unit ω caused by an interaction of the system with mechanism *v* during the time period [0,τ]. Moreover, define $A_{\omega }^{\mathrm{tot}}\left(v;\tau \right)=\int_{0}^{\tau }dtA_{\omega }\left(v;t\right)$ as the total (ensemble-average) number of state transitions in the unit ω that are caused by an interaction of the system with mechanism *v*. Then, using the positivity of the dynamical activity and of EP, together with the concavity of *g*, we can further bound the right-hand side to obtain a general limit for composite processes:(58)$$\begin{array}{c} l_{\omega }\leq \sum_{v:ℒ(v)\subseteq \omega }A_{\omega }^{\mathrm{tot}}\left(v;\tau \right)\;g\left(\frac{\zeta_{\omega }^{v}\left(\tau \right)}{A_{\omega }^{\mathrm{tot}}\left(v;\tau \right)}\right) \end{array}$$

This result provides an upper bound on how much $l_{\omega }$ can change during the time interval [0,τ], in terms of the associated activity of ω and the contribution of ω to EP. Thus, Equation (58) is a thermodynamic speed limit theorem, involving only the subsystems in the unit ω.

By comparison, the speed limit in [19] applied to a unit ω reads
(59)$$\begin{array}{c} l_{\omega }\leq A_{\omega }^{\mathrm{tot}}\left(\tau \right)\;f\left(\frac{\sigma_{\omega }\left(\tau \right)}{A_{\omega }^{\mathrm{tot}}\left(\tau \right)}\right) \end{array}$$For a composite process, the right-hand side of this “global” bound expands to
(60)$$\begin{array}{c} l_{\omega }\leq \left(\sum_{v:ℒ(v)\subseteq \omega }A_{\omega }^{\mathrm{tot}}\left(v;\tau \right)\right)\;\;g\left(\frac{\sum_{v:ℒ(v)\subseteq \omega }\zeta_{\omega }^{v}\left(\tau \right)}{\sum_{v:ℒ(v)\subseteq \omega }A_{\omega }^{\mathrm{tot}}\left(v;\tau \right)}\right) \end{array}$$

By Jensen’s inequality, the speed limit for composite processes (Equation (58)) is always tighter than the speed limit provided by [19] (Equation (59)). For a concave function *g*, a set of numbers $\left\{x_{v}\right\}$ in its domain, and positive weights $a_{v}$, Jensen’s inequality states that
(61)$$\begin{array}{c} \left(\sum_{v}a_{v}\right)g\left(\frac{\sum_{v}a_{v}x_{v}}{\sum_{v}a_{v}}\right)\geq \sum_{v}a_{v}g\left(x_{v}\right) \end{array}$$Setting $a_{v}=A_{\omega }^{\mathrm{tot}}\left(v;\tau \right)$ and $x_{v}=\frac{\zeta_{\omega }^{v}\left(\tau \right)}{A_{\omega }^{\mathrm{tot}}\left(v;\tau \right)}$ proves that Equation (58) is always tighter than Equation (59). Intuitively, this occurs because we are able to define the mechanism-specific contributions to the EP and activity in a composite process.

Ref. [19] provides some examples of acceptable functions *g*. For example, if we follow Pinsker’s inequality and choose $g=\sqrt{\frac{x}{2}}$, then the speed limit provided by [19] collapses to the speed limit derived in [17]. If we insert this choice of *g* into Equation (58), extract the parameter τ using the average frequency of state transitions ${\langle A_{\omega }\left(v\right)\rangle }_{\tau }=\frac{A_{\omega }^{\mathrm{tot}}\left(v;\tau \right)}{\tau }$, and rearrange the terms, we obtain the bounds
(62)$$\begin{array}{c} \forall \omega \in N^{†}:\;\tau \geq \frac{{\left(𝕃(p_{x_{\omega }}\left(0\right),p_{x_{\omega }}\left(\tau \right))\right)}^{2}}{2{\left(\sum_{v:ℒ(v)\subseteq \omega }\sqrt{\zeta_{\omega }^{v}\left(\tau \right){\langle A_{\omega }^{v}\rangle }_{\tau }}\right)}^{2}} \end{array}$$
the tightest of which is given by
(63)$$\begin{array}{c} \tau \geq \max_{\omega \in N^{†}}\frac{{\left(𝕃\left(p_{x_{\omega }}\left(0\right),p_{x_{\omega }}\left(\tau \right)\right)\right)}^{2}}{2{\left(\sum_{v:ℒ(v)\subseteq \omega }\sqrt{\zeta_{\omega }^{v}\left(\tau \right){\langle A_{}^{v}\rangle }_{\tau }}\right)}^{2}} \end{array}$$This particular speed limit tells us that the speed of the evolution of the system’s probability distribution cannot be greater than the speed of evolution of the distribution over the coordinates of the “slowest-evolving” unit.

## 6. Discussion

In this paper, we extend previous work on the stochastic thermodynamics of composite processes [4]. A central feature in the stochastic thermodynamics of a composite process is represented by the units of the process. These are subsets of the subsystems of the overall system whose joint dynamics does not depend on the state of the rest of the system. Our first contribution is to extend previous work that showed how the overlap among the units in a system allows us to “mix and match” previous bounds on EP from the literature (e.g., TURs, KURs, mismatch cost, SLTs, etc.) to derive strictly positive lower bounds on the EP by applying these bounds to the units within the system [2]. Crucially, as we show, this can sometimes be done when none of the previous bounds apply to the entire system as a whole.

We then present a preliminary analysis of how information flows in a composite process are constrained by the EPs of units. In the analysis, we demonstrate that bounds on the speed of transformation of a system’s probability distribution over states can be tightened with knowledge of the contributions to the EP and dynamical activity from each mechanism with which the system interacts.

This paper fits into a growing branch of research on the stochastic thermodynamics of constraints. One example of research in this area investigates the effect of constraints on the control protocol (time sequence of rate matrices evolving the probability distribution) [41]. There has also been some important work where the “constraint” on such a many-degree-of-freedom classical system is simply that it is some very narrowly defined type of system, whose dynamics is specified by many different types of parameters. For example, there has been analysis of the stochastic thermodynamics of chemical reaction networks [23,24], of electronic circuits [25,42,43], and of biological copying mechanisms [44]. This paper analyzes the consequences of a major class of dynamical constraints that arises because many of these systems are most naturally modeled as a set of multiple co-evolving subsystems [1,2,3,4,26,45,46,47]. In particular, the main constraints on such systems are that only certain subsets of subsystems can simultaneously change state a given time, and that the dependencies between subsystems impose restrictions on their joint dynamics.

There remain many avenues of potential future work, especially in the thermodynamics of computation. Many computational processes consist of multiple, co-evolving systems with a broad set of constraints that allow them to be easily modeled as a composite process. Moreover, the current physical implementations of (almost) all digital computations are periodic processes. Thus, the periodicity mismatch cost lower bound on EP in Equation (19) applies directly. Research in this direction would first require formalizing the notion of computation in a composite process. One such computation, which equates to the identity map, is simply communication (information transmission). One could extend the recent study on the fundamental thermodynamic costs of communication [48] to tie Shannon information theory to the stochastic thermodynamics of composite processes. More generally, for any given computation, one could analyze the trade-offs between the energy cost required to implement that computation and the performance (accuracy, time, etc.) of a composite process. In particular, there could be a rich structure in how the properties of the dependency network in a composite process affect these trade-offs.

We emphasize, though, that there are also issues for future work that do not involve physical implementations of computational systems. In particular, it is important to try to extend the composite systems’ TURs derived above for systems at an NESS to TURs with time-dependent driving [49] or discrete-time TURs [50]. (We thank an anonymous referee for this particular suggestion.)

## Figures and Tables

**Figure 1 entropy-25-01078-f001:**
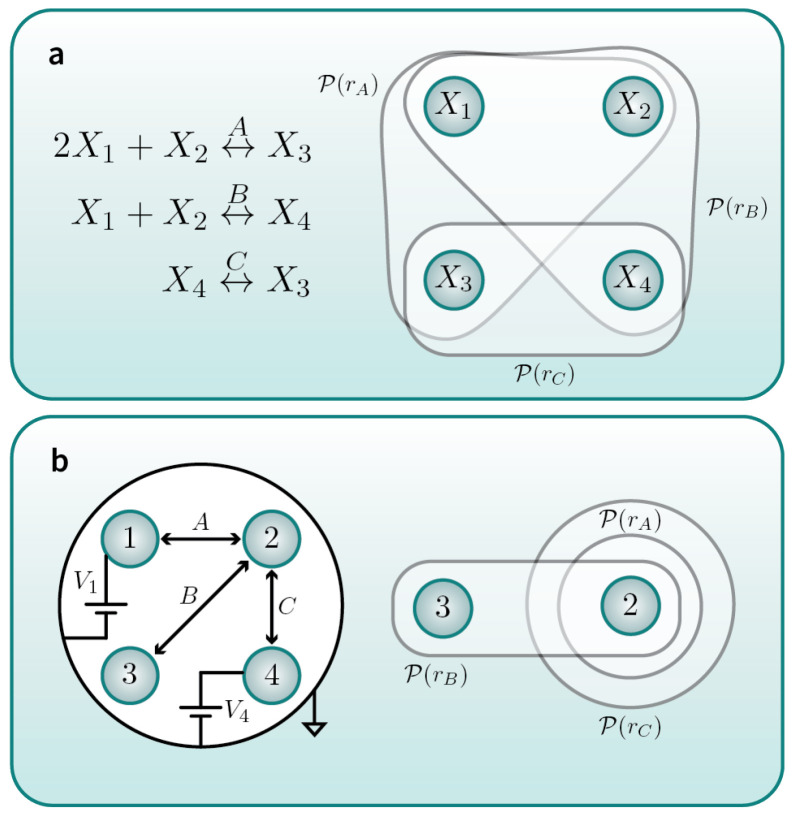
Examples of systems whose dynamics can be modeled as composite processes. Each system consists of multiple subsystems (blue circles). Mechanisms are denoted as *r*, and their puppet sets P(r) are indicated by translucent white bubbles. (**a**) An example stochastic CRN consists of four co-evolving species $\left\{X_{1},X_{2},X_{3},X_{4}\right\}$ that change state according to three chemical reactions {A,B,C}. (**b**) An example toy circuit consists of four conductors {1,2,3,4} that change state via interactions with three devices {A,B,C}.

**Figure 2 entropy-25-01078-f002:**
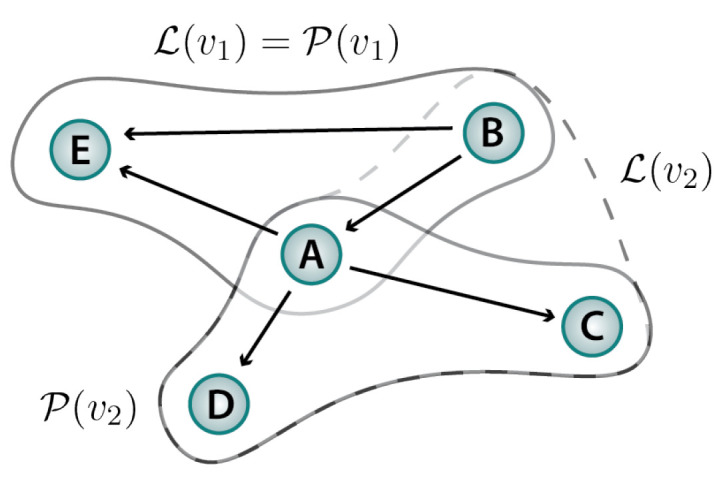
The dependency network specifies how the dynamics of each subsystem is governed by the states of other subsystems. This network defines the leader sets in a composite process.

## Data Availability

Not applicable.

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
