# Peer review of "Stochastic Thermodynamics of Multiple Co-Evolving Systems—Beyond Multipartite Processes"

_entropy, 2023, doi:10.3390/e25071078_

Round 1

Reviewer 1 Report

This paper fills a gap in the stochastic thermodynamics literature, in presenting a general formalism to handle composite systems (interconnected systems such as system of chemical reactions, electronic circuits  etc) in a way that allows to single out the contributions of each subsystem to the global entropy production, and write down various bounds (thermodynamic uncertainty relations for examples) in this formalism, which may lead to tighter bounds than if things are considered globally (ignoring the composite structure). Previous attempts to do so lacked generality to the point that they could not even handle simple, naturaly examples, such as two non trivial chemical reactions.

The paper is sound and interesting, and is a useful step to make stochastic thermodynamics more useful to analyse truly complex systems.

It is written for 'insiders': the basic concepts are not recalled. This can be understandable to some extent, but even the most basic notion (entropy production) is not recalled, which is a bit extreme. The symbol sigma-dot  appears in eq (7), and the symbol sigma (without dot) in eq (10). Even the acronym EP is used without explanation (it means 'entropy production'). Defining at least sigma and/or sigma dot at a general level (before the scary eq (14) ) would be nice. 

The paper is very formal (quite naturally, as it introduces a new formalism, so this is not a problem), leading to heavy notations sometimes. It leads to potentially interesting bounds, although no specific examples are given. Eg a small numerical (even if artificial) example of (40), and how it is better than (42), would help the reader understand the concepts by the illustration, as an alternative to the general notations. This is only a suggestion for better communication, not strictly required.

Another suggestion:  a name and interpretation for zeta^v_omega? 

To be clear: Does pa(i) include i  ? (in the leader set definition)  

A few typos: 

between eqs 40 et 41: unfinished sentence.

eq 22: missing ) 

Reviewer 2 Report

The authors of this study investigated the thermodynamic speed limit and thermodynamic uncertainty relations for multipartite Markov processes (MPP). The thermodynamic speed limit places a bound on the change of the system, as measured by the total variation distance, while taking into account both entropy production and dynamical activity. The thermodynamic uncertainty relation bounds the fluctuation of a given observable by the entropy production. The authors extended both of these concepts to multipartite processes, where no two subsystems can change state simultaneously, and derived improved bounds for the speed limit and the thermodynamic uncertainty relation by exploiting the dependence structure of MPP. These topics are of high interest in nonequilibrium physics, and the manuscript is well organized and written.

This manuscript substantially extends the previous preprint by the same authors (https://arxiv.org/abs/2107.12471), which focused solely on the thermodynamic speed limit. Compared to the preprint, the current manuscript has made significant improvements and is much more readable. However, it is unclear why the authors removed illuminating examples from their preprint. It would be beneficial if the authors included examples of the MPP bounds that demonstrate the effects of considering multi-partite structures.

Reviewer 3 Report

This manuscript studies thermodynamic bounds for composite systems. In particular, the authors find a new thermodynamic uncertainty relation and a new thermodynamic speed limit. The results are very interesting and, to the best of my knowledge, new. Therefore, I strongly recommend this manuscript for publication in entropy, after making the necessary changes.

There were some things that seemed wrong/unclear to me:
- At some points throughout the paper quantities are introduced without proper explanation. For example, the meaning of S in Eq. 9 is never given, the abbreviation EP is never properly introduced, and f is introduced in Eq. 27, but its meaning/constraints are only given in eq. 38. Similarly in Eq. 8, it is not clear to me what the definitions of N^*_1, N^*_2,... are. I suggest that the authors go through the manuscript again and make sure that everything is introduced consistently.
- Last equation on page 3: shouldn't '=' be ':'?
- In Eq. 18 under the first sum: why is the term x_{\omega}'=x_{\omega} omitted?
- Eq. 22: the bracket should be closed.
- The claim that 'Any positive monotonic concave function f relates the total variation distance to the KL divergence' seems wrong to me. For example, f(x)=\espilon*\sqrt(x), with \epsilon->0, would violate Eq. 38. The cited reference states that 'There exist various choices of a monotonic concave function' that satisfy the necessary property, but that is not the same as ANY monotonic concave function.
- As this manuscript focusses on thermodynamic uncertainty relations and speed limits, I do believe that it is appropriate to refer to the papers where these were originally introduced (i.e., refs. 1-2 below).

I also have some questions and optional remarks that are not strictly necessary but would in my opinion significantly strengthen the quality of the manuscript:
- I found the examples in Figures 1 and 2 very enlightning. I think it would increase the readability of the paper if the authors also illustrate the resulting TUR and TSL on these (or any other) examples.
- Eq. 7 seems quite important to the manuscript, so I think it would make the paper more self-contained if the authors could include an explicit proof of this relation (or at least a sketch of the basic ideas behind the proof).
- I was wondering whether it would be straightforward to extend the composite systems TUR derived in this manuscript to TURs with time-dependent driving (e.g. ref. 3 below) or discrete-time TURs (e.g. ref. 4 below)?

1 Barato, Andre C., and Udo Seifert. "Thermodynamic uncertainty relation for biomolecular processes." Physical review letters 114.15 (2015): 158101.
2 Aurell, Erik, Carlos Mejía-Monasterio, and Paolo Muratore-Ginanneschi. "Optimal protocols and optimal transport in stochastic thermodynamics." Physical review letters 106.25 (2011): 250601.
3 Koyuk, Timur, and Udo Seifert. "Thermodynamic uncertainty relation for time-dependent driving." Physical Review Letters 125.26 (2020): 260604.
4 Proesmans, Karel, and Christian Van den Broeck. "Discrete-time thermodynamic uncertainty relation." Europhysics Letters 119.2 (2017): 20001.

Round 2

Reviewer 3 Report

The authors have followed all of my recommendations.

There is, however, one minor thing that needs to be addressed: there seems to be something wrong with the reference before Eq. 13. I recommend accepting the manuscript for publication immediately after this is fixed.
